# Inhibitory Effects of Antipsychotic Chlorpromazine on the Survival, Reproduction and Population Growth Other Than Neurotransmitters of Zooplankton in Light of Global Warming

**DOI:** 10.3390/ijerph192316167

**Published:** 2022-12-02

**Authors:** Sen Feng, Yongzhi Zhang, Fan Gao, Meng Li, Lingyun Zhu, Hao Wen, Yilong Xi, Xianling Xiang

**Affiliations:** 1School of Ecology and Environment, Anhui Normal University, Wuhu 241002, China; 2Collaborative Innovation Center of Recovery and Reconstruction of Degraded Ecosystem in Wanjiang Basin Co-Founded by Anhui Province and Ministry of Education, Wuhu 241002, China

**Keywords:** psychoactive substances, global warming, zooplankton, dopamine

## Abstract

Global warming and environmental pollution have created a unique combination of abiotic and biotic stresses to zooplankton. However, little information is available on the effects of antipsychotic drugs commonly used to treat psychosis, such as chlorpromazine (CPZ), on non-target aquatic organisms in light of global warming. This study investigated how dopamine concentrations (DAC), acute toxicity and chronic toxicity of *Brachionus calyciflorus* changed in response to CPZ and gradually increasing temperatures. The results showed that the concentration range of rotifer DAC was 1.06~2.51 ng/g. At 18, 25 and 32 °C, the 24 h *LC_50_* was 1.795, 1.242 and 0.833 mg/L, respectively. Compared to the control, exposure to CPZ significantly decreased life expectancy at hatching, the net reproduction rate, generation time, population growth rate and dopamine concentration of *B. calyciflorus* in all three temperatures (*p* < 0.05). The toxicity of CPZ to rotifers was increased by high temperature. These findings indicated that CPZ is highly toxic to rotifers, displaying high ecological risks to aquatic ecosystems.

## 1. Introduction

Psychoactive substances (PSs) can alter mental functions such as thinking, emotion, and volitional action after being absorbed into the human body. With the increasing number of those suffering from mental illness and the global illicit drug trade, the demand for and use of PSs has risen rapidly in the last decade, and PS misuse has progressively become a global issue [1]. The unmetabolized PS products and inactive prodrugs enter municipal pipes with household wastewater after daily use [2]; however, these substances are not completely removed in the wastewater treatment plant (WWTPs), as some PSs and their metabolites are still released into natural water bodies, where they are deposited in sediments and accumulated in aquatic organisms, negatively influencing their behavior and the stability of aquatic ecosystems [3,4]. Currently, PSs have emerged as novel environmental pollutants with ecotoxic effects that have been discovered in a variety of aquatic environments, attracting significant attention [5].

Chlorpromazine (CPZ) is a low-cost phenothiazine psychoactive substance. As a first-generation antipsychotic, CPZ is commonly used to treat psychotic disorders such as delusions, hallucinations, mania, schizophrenia, and thinking disorders in humans [6]. CPZ reduces brain dopamine (DA) by inhibiting postsynaptic dopamine receptors (DARs) in the cerebral cortex and limbic regions, which would reduce psychotic symptoms such as delusions and hallucinations [7]. In addition, to its antipsychotic effects, CPZ has been considered among the most promising agents for inhibiting coronaviruses in humans (e.g., MERS-CoV and SARS-CoV-1) [8,9]. Based on the above findings, the hypothesis that CPZ could reduce the unfavorable evolution of COVID-19 and the infectivity of SARS-CoV-2 will be tested by a French team. It seems that inhibition of clathrin-mediated endocytosis is a critical factor in antiviral activity [10]. After 70 years of use and more than 25 years of research on residual drugs and their derivatives in the environment, CPZ is safe for everyday human use, but toxicological effects on aquatic organisms have also been reported for several non-target organisms, such as aquatic invertebrates [11,12], macrophytes [13], and fish [14]. Additionally, CPZ has been reported in various water bodies at concentrations ranging from 1 to 364 ng/L [15,16]. Even at trace levels, the unintentional presence of CPZ in aquatic ecosystems can cause devastating damage to organisms [17].

In the past decade, scholars have emphasized that climate change poses a direct or indirect threat to mental health [18,19]. Short-term extreme weather, long-term temperature rise and other climate changes caused by human activities are related to the deterioration of mental health, which inevitably increases the use of antipsychotics [20,21,22]. Generally, temperature affects the distribution and toxicity of environmental chemicals in water [23]. Convincing evidence shows that the increase in temperature due to global warming may cause irreversible damage to wildlife exposed to pollutants [24]. The rising water temperature may change the biological transformation of pollutants to more bioactive metabolites and impair homeostasis. Some researchers found that the toxicity of endosulfan to *Oncorhynchus mykiss* depends on temperature [25]. It was previously reported that the toxicity of chemicals all increased with higher temperature, such as pentachlorobenzene, parathion and chlorpyrifos to *Chironomus tentans* [26], fungicide chlorothalonil and insecticide Scourge^®^ to the estuarine grass shrimp *Palaemonetes pugio* [27]. However, the effects of increased temperature in light of global warming on the toxicity of antipsychotic drugs on zooplankton remain a daunting challenge.

Zooplankton occurs widely in freshwater ecosystems and is often used as a bioindicator to evaluate the potential health risks of exposure to trace elements and toxic chemical compounds [28,29]. Although *Daphnia magna* is among the most widely used zooplankton species in ecotoxicology, little is known about the toxicity of CPZ to this model system. It was found that the 48 h *EC_50_* of CPZ was 1.805 mg/L in *D. magna* [30], and its testing in toxicity assessment of another antipsychotic carbamazepine was reported [31]. Rotifers, especially *Brachionus calyciflorus*, *B. plicatilis* and *B. havanaensis*, have been widely used for toxicity assessment as model organisms because of their small size, ease of culture, rapid reproduction rate, and sensitivity to toxicity [32,33]. Additionally, the rotifer *B. calyciflorus* has a parthenogenetic reproduction mode, which provides a basis for us to test individuals with the same genetics [34]. Considering its higher sensitivity to drugs than other invertebrates, *B. calyciflorus* has been widely utilized to assess the potential toxicity of pharmaceuticals and their metabolites [35]; however, the toxic effects of CPZ remain poorly understood in rotifers.

We looked at the effects of CPZ on the neurotransmitter DA of *B. calyciflorus* as well as its life history and population growth parameters at high temperatures. The current study aims to (1) test the acute toxicity of CPZ to rotifers at different temperatures; (2) further understand the response of neurotransmitter DA in rotifers to CPZ toxicity at higher temperature; (3) determine the chronic effects of sublethal CPZ concentrations on rotifer development and reproduction at higher temperatures. The current investigation will reveal the toxic effects of CPZ on freshwater invertebrates in a systematic manner, which is critical for a comprehensive assessment of the environmental risk of antipsychotics to aquatic ecosystems.

## 2. Materials and Methods

### 2.1. Experimental Animals

Monoclonal cultures of *B. calyciflorus* were hatched from a single resting egg collected from the sediments of Lake Longwo (31°15′ N; 118°17′ E) in Wuhu city [36], and were identified morphologically and molecularly using a barcode (mitochondria COI sequence) as a biomarker [37]. Rotifers were cultured in EPA medium [38] at 25 °C with a 16:8 h (L:D) photoperiod and 1300 lx light intensity. Rotifers were fed *Tetradesmus obliquus* (Institute of Hydrobiology, Chinese Academy of Science, Wuhan, China) at a food density of 1.5 × 10^6^ cells/mL. Algae were grown in semi-continuous culture using HB-4 medium [39]. Rotifers were acclimated to temperatures of 18, 25 and 32 °C for over a week prior to the experiments to reduce interference from maternal effects [37].

### 2.2. Experimental Chemicals

Chlorpromazine hydrochloride (CAS: 69-09-0) was purchased from Sangon Biotech (Shanghai, China) with 98% of purity. The degradation of CPZ in the dark at room temperature was almost negligible: the half-life was estimated as 87.3 weeks. Meanwhile, the half-life of CPZ in darkness at 70 °C has been calculated as 4.55 weeks [40]. A stock solution of 5 mg/L was prepared by dissolving with distilled water to the desired concentration. We prepared new test solutions every 24 h to avoid contamination and light. All of the other reagents used were analytically pure.

### 2.3. Acute Toxicity Experiment

For acute toxicity assessment, seven concentrations (0 (control), 0.8, 1.2, 1.6, 2.0, 2.4, 2.8 and 3.2 mg/L) were included for each temperature (18, 25 and 32 °C). Ten rotifer juveniles (<4 h old) were chosen at random and placed in a glass jar containing 5 mL of test solution. Four replicates were performed for each treatment. Rotifers were cultured under darkness with 1.5 × 10^6^ cells/mL of *T. obliquus*. The number of dead rotifers was counted after 24 h, and the *LC_50_* was calculated using the Probit method [41].

### 2.4. Determination of DA Concentration

To explore the effects of CPZ on *B. calyciflorus* neurotransmitters in light of global warming, we set up groups of 0, 0.125, 0.25 and 0.5 mg/L CPZ treatments based on 24 h *LC_50_*, and measured changes in DA concentration (DAC). First, rotifers pre-cultured at 18, 25 and 32 °C were collected in three 2000 mL glass beakers according to temperature, and then rotifers in each temperature group were equally distributed to twelve 50 mL glass beakers. Each beaker held approximately 8000 rotifers and 50 mL of test solution. These beakers were kept at temperatures of 18, 25 and 32 °C for 12 h, with no food provided during this period. The rotifers were harvested by filtration through a 25 µm pore-sized mesh in a cryovial (washed with EPA medium and made up to 5 mL), rapidly frozen with liquid nitrogen and then stored at −80 °C for determination of DA levels. Each treatment was carried out three times.

Rotifers were homogenized for 10 min at 4 °C using a Bullet Blender Tissue Homogenizer (Next Advance, New York, NY, USA) at 2.0 m/s. The mixtures were then centrifuged at 3000× *g* for 10 min at room temperature. The supernatant was collected and subjected to assessment concentration using ELISA kits produced by Shanghai Elisa Biotech Co., Ltd. (Shanghai, China), following the manufacturer’s specifications. In brief, the DA level was determined using an enzyme-linked immunosorbent assay, and the absorbance at 450 nm was measured with an RT-6100 enzyme-labeled instrument (Rayto, Shenzhen, China).

### 2.5. Life Table Experiment

For the life table experiments, four CPZ concentrations (0 (control), 0.125, 0.25 and 0.5 mg/L) and three temperatures (18, 25 and 32 °C) were tested based on the 24 h *LC_50_*. Each treatment was replicated three times. In total, 36 glass jars (three temperatures × four CPZ concentrations × three replicates) were used. The experiment was conducted in 8 mL glass jars. For each glass jar, ten juveniles (<4 h old) were introduced to a 5 mL test solution containing 1.5 × 10^6^ cells/mL of *T. obliquus*. Every 24 h, 80% of the test solutions were replaced with freshly prepared solutions containing designated concentrations of algae. The cultured rotifers were checked every 12 h to record the numbers of neonates and dead individuals which were removed until all the experimental animals died.

Based on the data collected, age-specific survival (*l_x_*) and age-specific fecundity (*m_x_*) were calculated by using the conventional life table technique [42]. To drive the life table variables, including life expectancy (*e*_0_), generation time (*T*), net reproduction rate (*R*_0_) and intrinsic rate of population increase (*r_m_*) were calculated according to Krebs [43].

### 2.6. Population Growth Experiment

The temperature and CPZ concentration settings are the same as in Section 2.5. The experiments were carried out in 15 mL transparent test tubes, with 20 juveniles (<4 h old, hatched from amictic eggs under precultured conditions) placed in each test tube containing 10 mL test solution with 1.5 × 10^6^ cells/mL of *T. obliquus*. Each treatment was replicated three times. During the experimental, the number of live individuals in each replicate (using a whole sample count or 1~3 mL depending on population density) was counted daily and transferred into a freshly prepared test solution. *T. obliquus* deposited at the bottom of each test tube was resuspended with a micropipette every 12 h. The experiments were terminated when the population density peaked and continued to decline for 2~3 days.

The population growth rate (*r*) of the rotifers was calculated as follows:*r* = (ln *N_t_* − ln *N*_0_)/*t*(1)
where *t* is the culture day in which the rotifer density was the highest, and *N*_0_ and *N_t_* are the initial and highest rotifer densities, respectively [44].

### 2.7. Statistical Analysis

All data were expressed as the mean values with standard errors (mean ± SEs). The normality and homogeneity of data were tested using Kolmogorov–Smirnov’s tests and Levene’s tests, respectively. The 24 h *LC_50_* was derived using the Probit analysis. Two-way ANOVA was used to analyze the differences in life table parameters among all of the groups. Multiple comparisons were performed by the Student–Newman–Keuls q test (SNK-q test). A statistically significant difference was set as *p* < 0.05. All data were statistically analyzed using SPSS 22.0 (SPSS Inc., Chicago, IL, USA).

## 3. Results

### 3.1. Acute Toxicity Test

After 30 min of exposure, the *B. calyciflorus* reacted to each treatment by slowing down its swimming speed, forming clusters of two or three individuals, and weakening its response to external stimuli. The CPZ concentration at which rotifers began to die decreased from 1.2 to 0.4 mg/L as the temperature increased, and there was a significant linear relationship between CPZ concentration and rotifer mortality, respectively (Table 1). At 18, 25 and 32 °C, the 24 h *LC_50_* of CPZ to *B. calyciflorus* was 1.795, 1.242 and 0.833 mg/L, respectively. The 24 h *LC_50_* decreased with increasing temperature (Table 1), indicating that the toxicity of CPZ increased with temperature.

### 3.2. Changes in DAC

DAC in the rotifers of each treatment was calculated using the standard curve (Appendix A). At 25 °C, DAC in rotifers was highest (2.51 ng/g) in the control group, and lowest (1.06 ng/g) in the 0.5 mg/L CPZ treatments at 18 °C. DAC in rotifers decreased significantly with increasing CPZ concentration at all three temperatures (one-way ANOVA, all *p* < 0.05) when compared to controls. Temperature was significantly correlated with DAC in rotifers exposed to 0.125 mg/L CPZ treatments and controls, with DAC being highest at 25 °C (one-way ANOVA, *p* < 0.05, Figure 1).

CPZ concentration and temperature had a significant effect on rotifer DAC (two-way ANOVA, *p* < 0.01), but the CPZ × temperature interaction had no effect (two-way ANOVA, *p* > 0.05, Appendix A). There were significant nominal concentration–response relationships between CPZ concentration and DAC of *B. calyciflorus* at three different temperatures (Table 2).

### 3.3. Life Table Experiment

Using standard life table methods, we compared the *l_x_* and *m_x_* of rotifers exposed to different concentrations of CPZ at different temperatures. When compared to controls, the *l_x_* in all treatments tended to decrease earlier, and there was a trend toward a shorter time to death for all treatments for all individuals; treatments with CPZ at 0~0.5 mg/L significantly reduced the peak value of *m_x_* of 18 °C (one-way ANOVA, *p* < 0.05), but treatments with CPZ at 0.25 and 0~0.5 mg/L did not significantly affect the peak values of *m_x_* at 25 and 32 °C (one-way ANOVA, all *p* > 0.05, Figure 2a–f). Table 3 shows the results of selected life table variables of *B. calyciflorus* (18, 25 and 32 °C) exposed to different concentrations of CPZ. At 18, 25 and 32 °C, CPZ concentration had a significant effect on *e*_0_, *T*, *R*_0_ and *r_m_* (one-way ANOVA, all *p* < 0.05). Treatments with CPZ at 0.25~0.5, 0.25~0.5 and 0.125~0.5 mg/L significantly decreased the *e*_0_ at 18, 25 and 32 °C, respectively; treatments with CPZ at 0.5 mg/L significantly shortened *T* at 18, 25 and 32 °C, respectively; treatments with CPZ at 0.25~0.5 mg/L significantly decreased the *r_m_* values at 18 and 32 °C, and treatments with CPZ at 0.125 and 0.5 mg/L significantly decreased the *r_m_* values at 25 °C (one-way ANOVA, all *p* < 0.05). Except for treatments with CPZ at 0.125 mg/L at 32 °C and 0.25 mg/L at 25 °C, which were not significantly different from controls (one-way ANOVA, *p* > 0.05), CPZ significantly inhibited *R*_0_ in all treatments (one-way ANOVA, all *p* < 0.05). At 0~0.5 mg/L CPZ concentration, temperature had a significant effect on *e*_0_, *T*, *R*_0_ and *r_m_* (one-way ANOVA, all *p* < 0.05). Compared to controls, and at 18 °C, CPZ treatments of 0.5 and 0~0.5 mg/L significantly increased the *e*_0_ and *T* of rotifers, respectively, and the *R*_0_ and *r_m_* were significantly decreased in CPZ treatments of 0~0.5 mg/L (one-way ANOVA, all *p* < 0.05); similarly, at 32 °C, CPZ treatments of 0~0.5, 0~0.5 and 0.25~0.5 mg/L significantly decreased the *e*_0_, *T* and *R*_0_ of rotifers, respectively, and the *R*_0_ and *r_m_* were significantly increased in CPZ treatments of 0.25 and 0~0.5 mg/L (one-way ANOVA, all *p* < 0.05).

Appendix A displays the results of the two-way ANOVA performed on all the parameters of the rotifers subjected to CPZ concentration and temperature. Both CPZ concentration and temperature significantly affected *e*_0_, *R*_0_, and *T* of the rotifers (two-way ANOVA, all *p* < 0.001), but the CPZ × temperature interaction did not (two-way ANOVA, *p* > 0.05), and *r_m_* of the rotifers were significantly affected by CPZ concentration, temperature and their interaction (two-way ANOVA, all *p* < 0.05). At three temperatures of *B. calyciflorus,* significant nominal concentration–response relationships existed between CPZ concentration and *e*_0_, *T*, *R*_0_ and *r_m_* (Table 2).

### 3.4. Population Growth Experiment

The population growth curves of *B. calyciflorus* for each treatment are shown in Figure 3. The treatments with 0.5 mg/L CPZ at all temperatures demonstrated extremely unstable population dynamics, resulting in a low population density, and the treatments were terminated on day 6 when no live rotifers were discovered. This data set was only for display and not for analysis. At 18 °C, the growth rates of *B. calyciflorus* control and 0.125 mg/L CPZ treatments increased rapidly from day 3 and peaked on day 7 (78.63 ± 3.39 and 74.30 ± 2.66 ind./mL), whereas the population of 0.25 mg/L CPZ treatments grew slowly and peaked on day 11 (68.67 ± 1.76 ind./mL). Similarly, the *r* of rotifers decreased significantly as CPZ concentration increased (one-way ANOVA, all *p* < 0.05). When the temperature was increased to 25 °C, all treatments reached their highest peaks on days 11, 12 and 10, respectively (151.57 ± 4.79, 121.93 ± 4.30 and 131.13 ± 1.68 ind./mL), but the *r* of rotifers significantly decreased in comparison to the control (one-way ANOVA, all *p* < 0.01). At 32 °C, both treatments with CPZ at 0.125 mg/L and the control reached the peak on day 12 (103.33 ± 5.29 and 116.20 ± 4.39 ind./mL), and treatments with CPZ at 0.25 mg/L reached a peak on day 13 (91.17 ± 2.60 ind./mL), the *r* of rotifers exposed to 0.25 mg/L CPZ was significantly lower than the controls (one-way ANOVA, *p* < 0.05, Table 4). The maximum population density was significantly decreased at 18 and 32 °C (one-way ANOVA, all *p* < 0.05); the *r* was significantly lower at 18 °C only in the 0.25 mg/L CPZ treatments, with no significant differences between the other treatments compared to the controls.

## 4. Discussion

The first clinical application of chlorpromazine in France in 1952 was considered to be the beginning of psychopharmacology [45]. Since its birth 70 years ago, the use of antipsychotic drugs has continued to increase, and the scope of use has expanded to companion animals. Although antipsychotic drugs pose an important threat to the environment to varying degrees [46], people have only begun to pay attention to antipsychotic drugs in the environment in recent decades [47]. This has led to the lack of appropriate environmental risk assessments for most antipsychotic drugs. Here, we take *B. calyciflorus* as the test organism to study the survival, population growth and dopamine concentration of rotifers under CPZ and temperature stress, in an attempt to provide a basis for environmental risk assessment of CPZ.

Acute toxicity of CPZ to aquatic organisms has been documented. The acute 24 h *LC_50_* obtained for CPZ (0.833–1.795 mg/L) with *B. calyciflorus* is higher than those obtained for aquatic organisms such as the protozoan *Spirostomum ambiguum* (24 h *EC_50_* of 0.35 mg/L), the freshwater goldfish *Carassius auratus* (48 h *EC_50_* of 0.43 mg/L) and the crustaceans *Thamnocephalus platyurus* (24 h *LC_50_* of 0.62 mg/L), similar to the *D. magna* (48 h *EC_50_* of 1.805 mg/L) [14,30,48]. Other studies on CPZ ecotoxicity are difficult to compare with our findings because the experimental endpoints differ greatly [49,50,51]. In conclusion, the sensitivity of *B. calyciflorus* to CPZ is comparable to that of *D. magna*, but more sensitive than that of fish and protozoa. Some researchers investigated the approximate lethal doses of 58 compounds at different temperatures in rats and discovered that the majority of the compounds were 2–17 times more toxic at 36 °C than at 26 or 8 °C [52]. Interestingly, CPZ, promazine and strychnine are more toxic at 8 and 36 °C, but much less toxic at room temperature. Our findings support the majority of compounds, with CPZ being more toxic to *B. calyciflorus* at 32 °C than at 25 or 18 °C.

The DA system was prevalent in both vertebrates and invertebrates and performs more of a neurotransmitter function. In vertebrates, DA was involved in many functions, including locomotion [53,54,55], cognition [56,57,58] and development [59,60]. In each temperature group of this experiment, the DAC of rotifers decreased significantly with the increase in CPZ concentration, and there was a concentration–response relationship between CPZ concentration and DAC. While there were no appreciable variations in DAC at the three temperatures in the higher concentration treatments, there was a tendency for DAC in the lower CPZ treatments for all samples to decline sequentially at 25, 32, and 18 °C. Increasing CPZ concentrations had a clear negative effect on rotifer DAC at all temperatures; however, the relationship between rotifer DAC and rotifer life history and population growth has not been reported. In conclusion, the DA system may be important in regulating life history and population growth in response to exogenous stress. Chemically induced changes in DA levels may alter rotifer life history responses, but the precise effects must be investigated.

Life table experiments on rotifers are common in environmental risk assessment, ecological and aquaculture research [61]. At all temperatures, there was no significant difference in *T* at 0.125 mg/L CPZ treatments compared to the control in the current study. Previous research had shown that low toxicant exposure improved performance, a phenomenon known as hormesis [62]. The *e*_0_, *R*_0_ and *r_m_* were significantly reduced in treatments with 0.25 mg/L CPZ at 18 and 32 °C. Consistently, the *e*_0_, *R*_0_, *T* and *r_m_* were significantly lower in treatments with 0.5 mg/L CPZ than in the control at all temperatures. Higher concentrations of CPZ had more toxic effects on *e*_0_, *R*_0_, *T* and *r_m_* were obvious in all three temperatures, which was consistent with previous studies [63,64]. With increasing temperatures, the *T* shortened significantly and the *r_m_* increased significantly in all treatments. In our experiments, we discovered that rotifers became significantly smaller as the temperature rose, so we hypothesized that rotifers were allocating more energy to reproduction rather than growth and development, which was also consistent with previous research [65]. Two-way ANOVA revealed significant interactive effects between temperature and CPZ concentration on *r_m_* of rotifers, implying that temperature may affect CPZ toxicity. More specifically, when compared to the control, *e*_0_ in 0.125 mg/L CPZ treatments with did not change at 18 °C but significantly decreased at 32 °C. Similarly, *R*_0_ in 0.125 mg/L CPZ treatments did not change at 25 °C but significantly decreased at 32 °C when compared to the control. These findings suggested that high temperatures may increase CPZ toxicity to rotifers. Similarly, the toxicity of nTiO_2_ and imidacloprid to *B. calyciflorus* with increasing temperature [66,67], as well as the toxicity of terbufos, trichlorfon, and imidacloprid to *Palaemonetes spp.*, *Cyprinodon variegatus* and *Isonychia bicolor* [68,69], and the toxicity of an ethoprop- malathion mixture to *Oncorhynchus kisutch* [70]. In line with our findings, the toxicity of CPZ to zooplankton is increasing with global warming.

So far, adverse effects on population demographic factors such as *r* have been widely used as reliable indicators to demonstrate reliable various environmental stressors such as chlordecone, fipronil, chloramphenicol and oxytetracycline in the genus *Brachionus* (e.g., *B. plicatilis* and *B. calyciflorus*) [71,72,73]. When organisms are subjected to external stresses, the majority of their energy is directed toward mitigating the effects of the stressor, sacrificing energy for growth and reproduction [74]. For zooplankton, population density and *r* are critical at the trophic level [71]. In this experiment, 0.125 mg/L CPZ significantly increased maximum population density at 25 °C, while 0.25 mg/L CPZ significantly decreased maximum population density at 18 and 32 °C. We discovered more rotifer corpses during the count, indicating that the rotifer population was in a collapsed stage at the time of the count, and the maximum population density was affected by the experimental container and the experimental volume [75]. CPZ had a negative effect on the *r* at all temperatures. On the sixth day, population growth was slowed and extinction was particularly observed in *B. calyciflorus* exposed to 0.5 mg/L CPZ. The effects of high CPZ concentrations on rotifer population dynamics could be explained by the following factors. To begin, the swimming speed of the rotifer determines its water filtration rate and feeding rate [76], and rotifers move significantly slower in highly concentrated CPZ solutions, where food deprivation may be among the main factors influencing population growth. Secondly, the activities of several enzymes, including lipase, amylase and acetylcholinesterase, as well as the expression of cytochrome P450s genes, have been linked to changes in rotifer populations [67], but the effect of CPZ on rotifer survival and reproduction has yet to be determined. Finally, as a classic antipsychotic, CPZ inhibits DAR in organism cell membranes, interfering with the neuroendocrine system and affecting survival and population growth. Overall, changes in rotifer life history and population growth parameters may be attributed to altered rotifer DAC in this study.

## 5. Conclusions

According to the findings of this study, CPZ was highly toxic to *B. calyciflorus*, and the toxicity increased with increasing temperature within a certain range. Dopamine concentrations in rotifers decreased significantly as CPZ concentrations increased at different temperatures. The life expectancy, net reproduction rate, generation time, intrinsic growth rate and population growth rate of *B. calyciflorus* were all significantly affected by different temperatures and CPZ concentrations. Overall, CPZ pollution may not be acutely toxic to rotifers, but it may still have an impact on rotifers in light of global warming, which may then influence community structure in natural waters.

## Figures and Tables

**Figure 1 ijerph-19-16167-f001:**
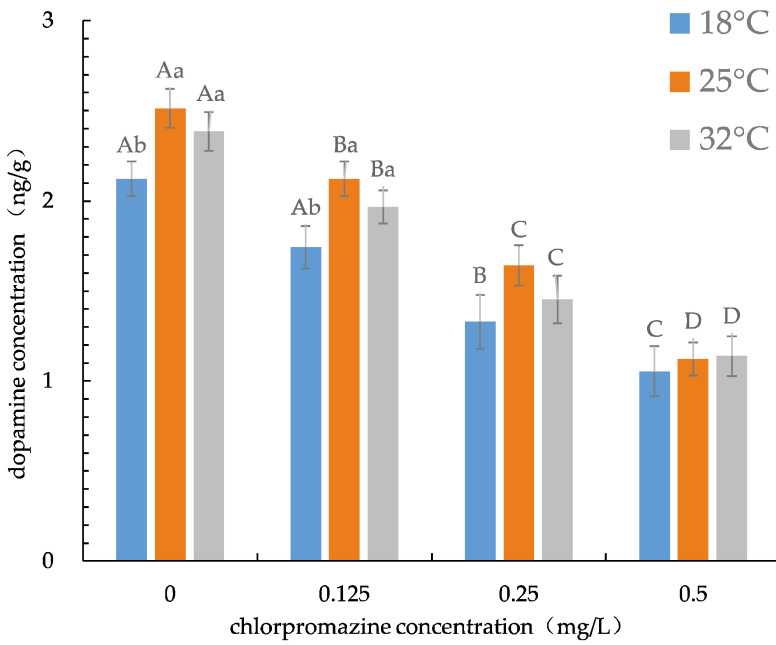
Dopamine concentration of rotifers at different chlorpromazine concentrations and temperatures. Values in each column with different capital letters indicate significant differences among treatments for each temperature (*p* < 0.05); column values with different small letters indicate significant differences among treatments for each chlorpromazine hydrochloride concentration (*p* < 0.05).

**Figure 2 ijerph-19-16167-f002:**
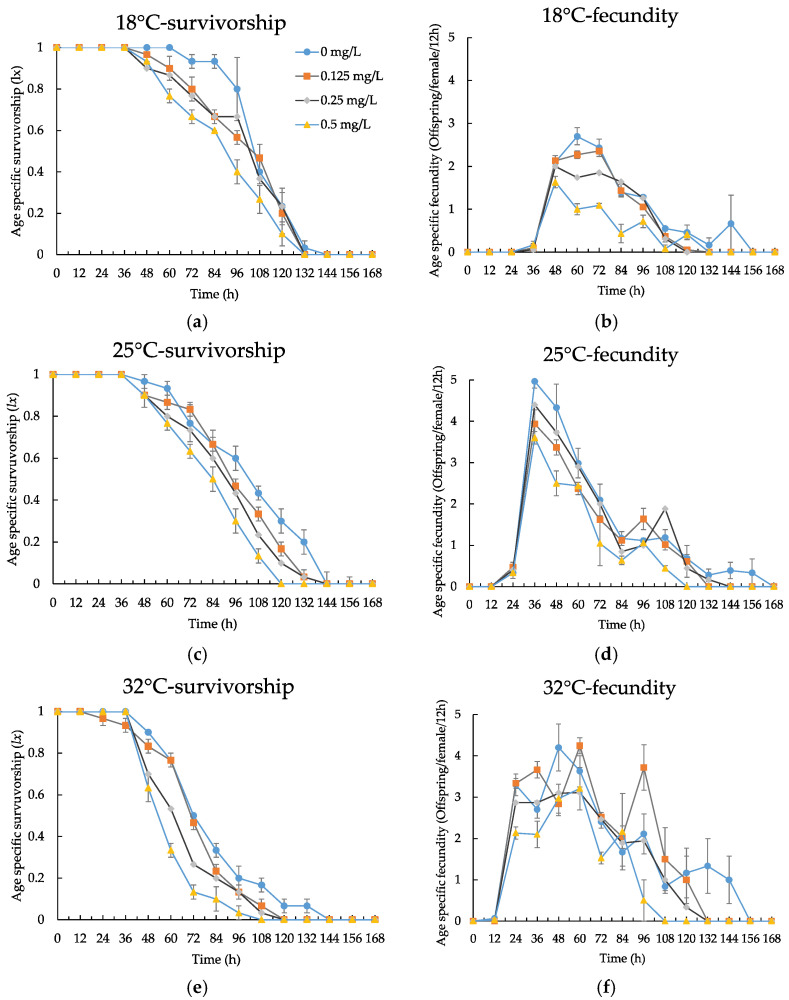
Age-specific survivorship and age-specific fecundity of *B. calyciflorus* exposed to different concentrations of chlorpromazine at different temperatures. Shown are the mean ± standard error based on three replicates for: (**a**) 18 °C—survivorship; (**b**) 18 °C—fecundity; (**c**) 25 °C—survivorship; (**d**) 25 °C—fecundity; (**e**) 32 °C—survivorship; (**f**) 32 °C—fecundity.

**Figure 3 ijerph-19-16167-f003:**
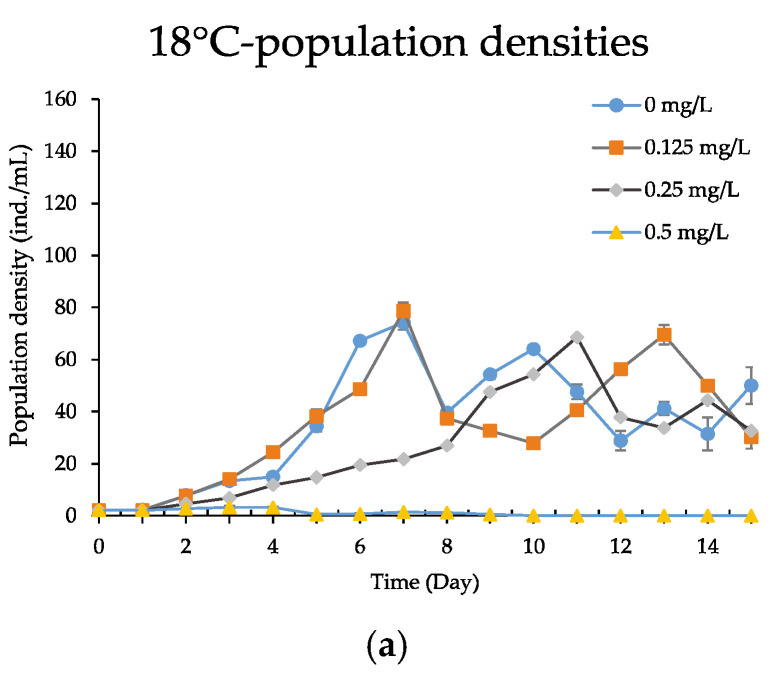
Population growth curves expressed by population densities of the *B. calyciflorus* exposed to chlorpromazine cultured at 18, 25 and 32 °C. Shown are the mean ± standard error based on three replicates for: (**a**) 18 °C—population densities; (**b**) 25 °C—population densities; (**c**) 32 °C—population densities.

**Table 1 ijerph-19-16167-t001:** The 24 h *LC_50_* of chlorpromazine as well as the relationships between chlorpromazine concentration (*x*, mg/L) and mortality rate of *B. calyciflorus* at three temperatures. Shown are the mean ± standard error based on three replicates.

Temperature (°C)	24 h *LC_50_* (mg/L)	95% Confidence Limits	Regressive Equations	Significant Tests
18	1.795	1.669~1.918	*y* = 0.375*x* − 0.178	*R*^2^ = 0.907, *p* < 0.001
25	1.242	0.795~1.644	*y* = 0.390*x* − 0.054	*R*^2^ = 0.847, *p* < 0.001
32	0.833	0.728~0.933	*y* = 0.3412*x* + 0.142	*R*^2^ = 0.795, *p* < 0.001

**Table 2 ijerph-19-16167-t002:** The relationships between chlorpromazine concentration (*x*, mg/L) and life expectancy at hatching (*e*_0_), generation time (*T*), net reproductive rate (*R*_0_), intrinsic rate of population increase (*r_m_*) and dopamine concentration (*DAC*) of *B. calyciflorus* at three temperatures.

Temperature (°C)	Parameters	Regressive Equations	Significant Tests
18	*e* _0_	*y* = 67.491*x*^2^ − 71.156*x* + 105.629	*R*^2^ = 0.744, *p* < 0.01
	*R* _0_	*y* = 8.000*x*^2^ − 16.120*x* + 9.837	*R*^2^ = 0.953, *p* < 0.001
	*T*	*y* = −34.764*x*^2^ + 5.332*x* + 68.094	*R*^2^ = 0.696, *p* < 0.01
	*r_m_*	*y* = 0.00034*x*^2^ − 0.027*x* + 0.036	*R*^2^ = 0.903, *p* < 0.001
	*DAC*	*y* = 14.749*x*^2^ − 16.110*x* + 8.564	*R*^2^ = 0.831, *p* < 0.001
25	*e* _0_	*y* = 59.927*x*^2^ − 71.811*x* + 101.793	*R*^2^ = 0.786, *p* < 0.001
	*R* _0_	*y* = 5.479*x*^2^ − 15.752*x* + 16.005	*R*^2^ = 0.686, *p* < 0.01
	*T*	*y* = −17.745*x*^2^ − 2.422*x* + 55.252	*R*^2^ = 0.621, *p* < 0.05
	*r_m_*	*y* = −0.005*x*^2^ − 0.014*x* + 0.059	*R*^2^ = 0.657, *p* < 0.01
	*DAC*	*y* = 9.202*x*^2^ − 15.940*x* + 10.134	*R*^2^ = 0.926, *p* < 0.001
32	*e* _0_	*y* = 45.964*x*^2^ − 65.585*x* + 78.076	*R*^2^ = 0.925, *p* < 0.001
	*R* _0_	*y* = −1.552*x*^2^ − 15.059*x* + 15.538	*R*^2^ = 0.909, *p* < 0.001
	*T*	*y* = 6.012*x*^2^ − 18.352*x* + 49.255	*R*^2^ = 0.618, *p* < 0.05
	*r_m_*	*y* = −0.039*x*^2^ − 0.011*x* + 0.072	*R*^2^ = 0.905, *p* < 0.001
	*DAC*	*y* = 16.931*x*^2^ − 18.730*x* + 9.652	*R*^2^ = 0.893, *p* < 0.001

**Table 3 ijerph-19-16167-t003:** Effects of temperatures and chlorpromazine on the life history demographic parameters of *B. calyciflorus*. Shown are the mean ± standard error based on three replicates.

Parameters	CPZ Concen. (mg/L)	Temperature (°C)		
		18	25	32
Life expectancyat hatching (h)	0	106.0 ± 4.2 ^Aa^	102.0 ± 4.8 ^Aa^	78.0 ± 2.1 ^Ab^
	0.125	96.8 ± 2.8 ^ABa^	93.2 ± 1.7 ^ABa^	70.8 ± 0.7 ^Bb^
	0.25	92.8 ± 2.6 ^BCa^	88.0 ± 0.8 ^BCa^	64.4 ± 2.2 ^Cb^
	0.5	86.8 ± 0.4 ^Ca^	80.8 ± 2.2 ^Cb^	56.8 ± 0.4 ^Dc^
Generationtime (h)	0	68.8 ± 1.0 ^Aa^	55.1 ± 1.6 ^Ab^	49.0 ± 2.3 ^Ac^
	0.125	66.3 ± 0.4 ^Aa^	55.2 ± 0.4 ^Ab^	47.7 ±0.9 ^Ac^
	0.25	68.7 ± 0.9 ^Aa^	53.2 ± 1.2 ^ABb^	44.6 ± 1.8 ^ABc^
	0.5	61.9 ± 0.8 ^Ba^	49.7 ± 1.2 ^Bb^	41.7 ± 0.3 ^Bc^
Net reproductiverate	0	9.9 ± 0.2 ^Ab^	16.5 ± 1.6 ^Aa^	15.2 ± 0.7 ^Aa^
	0.125	7.9 ± 0.3 ^Bc^	12.8 ± 0.5 ^Bb^	14.5 ± 0.3 ^Aa^
	0.25	6.4 ± 0.4 ^Cc^	13.4 ± 0.9 ^ABa^	11.0 ± 0.2 ^Bb^
	0.5	3.8 ± 0.3 ^Db^	9.4 ± 0.6 ^Ca^	7.7 ± 0.6 ^Ca^
Intrinsic rate of population increase (h)	0	0.0357 ± 0.0005 ^Ac^	0.0598 ± 0.0014 ^Ab^	0.0714 ± 0.0016 ^Aa^
	0.125	0.0330 ± 0.0010 ^Ac^	0.0549 ± 0.0014 ^Bb^	0.0705 ± 0.0003 ^Aa^
	0.25	0.0288 ± 0.0011 ^Bc^	0.0568 ± 0.0007 ^ABb^	0.0660 ± 0.0010 ^Ba^
	0.5	0.0225 ± 0.0017 ^Cc^	0.0504 ± 0.0011 ^Cb^	0.0566 ± 0.0017 ^Ca^

Different capital letters indicate a significant difference in each column and different lowercase letters indicate a significant difference in each row for each parameter (*p* < 0.05).

**Table 4 ijerph-19-16167-t004:** Population parameters of *B. calyciflorus* to three concentrations of chlorpromazine at different temperatures. Shown are the mean ± standard error based on three replicates.

Temperature (℃)	CPZ Concen. (mg/L)	Parameters		
		Day of Maximum Density	Maximum Rotifer Density (ind./mL)	Population Growth RATE
18	0	7	74.30 ± 2.66	0.40 ± 0.02
	0.125	7	78.63 ± 3.39	0.29 ± 0.02 **
	0.25	12	68.67 ± 1.76 ***	0.23 ± 0.01 ***
	0.5	4	3.30 ± 0.49	0.12 ± 0.04
25	0	10	131.13 ± 1.68	0.57 ± 0.03
	0.125	11	151.57 ± 4.79 **	0.42 ± 0.04 **
	0.25	12	121.93 ± 4.30	0.38 ± 0.01 **
	0.5	4	16.17 ± 1.13	0.20 ± 0.01
32	0	12	116.20 ± 4.39	0.67 ± 0.08
	0.125	12	103.33 ± 5.29	0.48 ± 0.07
	0.25	13	91.17 ± 2.60 **	0.41 ± 0.01 *
	0.5	3	17.77 ± 0.62	0.35 ± 0.02

* Significant difference from the control. * *p* < 0.05, ** *p* < 0.01, and *** *p* < 0.001.

## Data Availability

The data showed in this study are available on request from the corresponding author.

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
