# Peer review of "Inhibitory Effects of Antipsychotic Chlorpromazine on the Survival, Reproduction and Population Growth Other Than Neurotransmitters of Zooplankton in Light of Global Warming"

_ijerph, 2022, doi:10.3390/ijerph192316167_

Round 1
Reviewer 1 Report
This study examines how antipsychotic chlorpromazine affects the survival, reproduction and population growth other than neurotransmitter of Brachionus calyciflorus in response to global warming. While it is interesting I think the statistical that lacks some proper validation on whether the temperature affects the response variables, such as growth and life expectancy. The authors should consider the use of mixed-effect models, which are ideal for such a kind of data that represent kinetics over time. Apart from this major issue, I have some small comments.
Line 31” they have been treated eventually by sewage treatment facilities, some PSs and their”. Treating might be an appropriate word for water and wastewater, but I think the substances are not treated but degraded/removed/metabolized during treatment.
Line 44 Inadequate reference for coronavirus (SARS-CoV-2).
Line 51 Transition to climate change comes out of nowhere. The author should improve the transition.
Line 111 What is the “probit” method? The authors should elaborate.
Line 132 The use words instead of digits should be preferred for numbers between 1 to 12.
Line 137 Informal language such as “test solutions were changed”, should be turned to more formal language: e.g. “test solutions were replaced”.
Line 144 Bad citation, where is the date, was the author all alone?
Line 173. When report p-values, tests should be reported as well. Also what the “all” does mean. The authors should elaborate.
Line 182. The figure 1(a) should go to supplementary material
Please add a title on each plot in Figure 2. It will significantly help the peer readers.
Line 254 What ”ind./mL” means? The authors should elaborate.
Line 276 Very bad introduction to discussion, which does not add anything to the discussion section. The authors should improve the transition.
Lines 276-281 Does the comparison with tests for other organisms conducted by different studies, relate to the results? Should the authors expect also biases due to the tests being performed in other studies from different groups?
Reviewer 2 Report
The experiment is surely well described and performed, however, the language used often tends to over-emphasize the meaning of the work. For example, at lines 61-64 it seems that the present project is imperative for the sake of environment, where it does not have much ecological relevance. As reported by the authors, the environmental concentration of Chlorpromazine ranges from 1 to 364 ng/L, and the authors used concentrations 1000 times higher. Moreover, Chlorpromazine has a half-life of a few hours/days, an important detail not mentioned by the authors.
Are the temperature used environmentally relevant for the species used and in a climate change scenario?
Did the authors measure the actual concentration of the CPZ on stock solution/working solution/exposure medium?
Lines 115-117. “Rotifers were pre-cultured at 18, 25 and 32°C, pooled in a 2000-mL glass beaker and then equally distributed into 12 50-mL glass beakers.” Does it mean that rotifers from different temperatures were pooled together? It is not clear.
Grammar errors can be found throughout the text, for example check the verb at line 290, why did the author used “was”?
